# Ligand Binding Properties of Odorant-Binding Protein OBP5 from *Mus musculus*

**DOI:** 10.3390/biology12010002

**Published:** 2022-12-20

**Authors:** Lucie Moitrier, Christine Belloir, Maxence Lalis, Yanxia Hou, Jérémie Topin, Loïc Briand

**Affiliations:** 1Centre des Sciences du Goût et de l’Alimentation, CNRS, INRAE, Institut Agro, Université Bourgogne Franche-Comté, F-21000 Dijon, France; 2Institut de Chimie de Nice UMR7272, Université Côte d’Azur, CNRS, 28 Avenue Valrose, 06108 Nice, France; 3Université Grenoble Alpes, CEA, CNRS, INAC-SyMMES, 17 Rue des Martyrs, 38000 Grenoble, France

**Keywords:** odorant-binding protein, lipocalin, purification, isothermal microcalorimetry, affinity, fluorescent assay, circular dichroism, odorant

## Abstract

**Simple Summary:**

Odorant-binding proteins are soluble proteins abundantly secreted in the nasal mucus of vertebrates. Although their physiological functions are not known, these proteins are suspected to have a carrier role in solubilizing and carrying hydrophobic odorant molecules to the olfactory receptors. Here, we describe the expression of functional mouse mOBP5. The protein produced using bacteria was purified and characterized. We investigated its binding properties using a fluorescent competitive assay and microcalorimetry. Molecular docking experiments revealed hydrophobic residues in the binding cavity potentially involved in the stabilization of the odorant, thus explaining its binding properties.

**Abstract:**

Odorant-binding proteins (OBPs) are abundant soluble proteins secreted in the nasal mucus of a variety of species that are believed to be involved in the transport of odorants toward olfactory receptors. In this study, we report the functional characterization of mouse OBP5 (mOBP5). mOBP5 was recombinantly expressed as a hexahistidine-tagged protein in bacteria and purified using metal affinity chromatography. The oligomeric state and secondary structure composition of mOBP5 were investigated using gel filtration and circular dichroism spectroscopy. Fluorescent experiments revealed that mOBP5 interacts with the fluorescent probe *N*-phenyl naphthylamine (NPN) with micromolar affinity. Competitive binding experiments with 40 odorants indicated that mOBP5 binds a restricted number of odorants with good affinity. Isothermal titration calorimetry (ITC) confirmed that mOBP5 binds these compounds with association constants in the low micromolar range. Finally, protein homology modeling and molecular docking analysis indicated the amino acid residues of mOBP5 that determine its binding properties.

## 1. Introduction

Vertebrate odorant-binding proteins (OBPs) are small (~20 kDa) soluble proteins highly secreted in the nasal mucus. OBPs reversibly bind a wide range of odorant molecules with affinities in the micromolar range [1,2,3]. Although the physiological function of vertebrate OBPs is not clearly established, it is broadly accepted that OBPs solubilize airborne odorants, which are commonly hydrophobic molecules, thus serving as carriers toward olfactory receptors across the hydrophilic nasal mucus [1,3]. Recently, OBP scavenging capabilities for several signaling molecules secreted by different pathogenic microorganisms have been demonstrated, suggesting antimicrobial and fungistatic activity for vertebrate OBPs [4].

Vertebrate OBPs belong to the lipocalin protein family. They have low sequence similarities, but share a conserved three-dimensional structure made of an eight-stranded anti-parallel beta-barrel domain connected to a short carboxy-terminal alpha-helix [5]. The beta-barrel forms an apolar pocket called the calyx, whose function is to bind hydrophobic molecules, such as odorants [6]. Another group of lipocalins involved in chemical communication are major urinary proteins (MUPs), which are generated by the liver and secreted in the urine [7]. However, MUPs have been reported to be expressed in other tissues, such as mammary, salivary, olfactory, and lacrimal glands [8,9].

Vertebrate OBPs have been observed in a large variety of mammalian species, such as cows, pigs, xenopus, rabbits, rats, goats, sheep, elephants, and humans [1,2,3,10,11,12,13]. A few different types of OBPs have been found in a variety of animal species. Four OBPs have been observed in the nasal mucus of the giant panda [14], and three OBPs have been reported in the nasal mucus of rat, which were shown to be complementary in odorant binding properties [15,16,17]. Proteomic analysis has revealed different OBP isoforms in pigs, giant pandas, goats, and sheep originating from phosphorylation and glycosylation [11,14,18,19]. The comparison of native and recombinant pig OBP isoforms revealed that post-translational modifications (PTMs) affect the different binding affinities of OBPs for specific ligands [18,20].

In mice, proteomic and transcriptomic analyses have revealed the presence of four highly expressed OBPs (OBP1, OBP2, OBP5, and OBP7) [21,22,23,24,25]. At the RNA level, OBP2, OBP1, and OBP5 are the most strongly expressed OBPs in the main olfactory epithelium [22], which is in agreement with protein isolation from mouse nasal tissue [24]. In situ hybridization experiments have demonstrated that genes encoding mouse OBP5 and OBP7 are expressed in the septal and lateral nasal glands, but not in glands from the olfactory mucosa [24,26]. Although the functional role of OBPs is not clearly understood, using a fluorescently labeled mOBP5 loaded with an odorant, Strotmann and Breer have showed a rapid internalization of OBP inside the sustentacular cells of the mouse epithelium [27].

In addition, two other groups of lipocalins have been shown to be expressed in the main olfactory epithelium (LCN3, LCN4, MUP4, and MUP5) [9,22]. The role of these lipocalins in olfaction is not known, but their capability to bind hydrophobic molecules suggests that they may widen the spectrum of odorant or pheromone molecules carried by OBPs [28,29].

Because vertebrate OBPs are remarkably stable, relatively easy to express, and have the desired binding properties that can be modified through site-directed mutagenesis and molecular modeling [30,31], they have great potential for biotechnological applications [32,33]. OBPs can be used for the development of sensing elements in biosensors for odor monitoring, clinical analysis, or food sensory evaluation [34,35,36,37,38,39]. OBPs can also be an efficient solution to prevent and/or remove unpleasant odorant molecules trapped on the surface of textiles [40].

Here, we report the expression of functional mouse mOBP5 using *Escherichia coli*, which is one of the most abundantly expressed OBPs in olfactory tissues of the house mouse [22,24]. The recombinant protein was purified and characterized for its secondary and quaternary structures. We also investigated its binding properties for several odorants using a fluorescent competitive assay and microcalorimetry. As the recognition spectrum of this OBP is unknown, molecules from different chemical families with different impact on behavior were selected [41]. Molecular docking results showed that hydrophobic residues in the binding cavity are strongly involved in the stabilization of the odorant. In addition, a polar residue was identified as having a major role in the orientation of the odorant within the binding cavity. Overall, our results suggest that binding to mOBP5 is primarily controlled by the size of the odorant, independent of its chemical function.

## 2. Materials and Methods

### 2.1. mOBP5 Production and Purification

Recombinant mOBP5 was produced as previously described [42]. Briefly, the cDNA sequence encoding *Mus musculus* mOBP5 (formerly known as mOBP1a, UniProt ID: Q9D3H2) minus its signal sequence was synthesized using Genewiz (Leipzig, Germany) with codon usage optimized for expression in *E. coli*. The synthetic DNA of mOBP5 was subcloned into the *BamH*I and *Hind*III restriction sites of the pQE31 plasmid (Qiagen, Courtaboeuf, France). The resulting expression plasmid pQE31-mOBP5 encodes a fusion protein (Appendix A) composed of an N-terminal His_6_-tag, followed by mOBP5 (Ala17-Glu163). The plasmid was subsequently introduced into *E. coli* M15 [pREP4] cells (pQE expression system, Qiagen). The expression of mOBP5 was induced with 0.2 mM isopropyl β-D-1-thiogalactopyranoside (IPTG) for 3 h at 37 °C in Terrific Broth medium supplemented with 100 µg/mL ampicillin and 25 µg/mL kanamycin.

Recombinant His-tagged mOBP5 protein was purified as previously described [42] using a HisTrap^TM^ Ni^2+^-chelating column (GE Healthcare). To remove potentially bound contaminants, the purified mOBP5 protein was extensively dialyzed for 24 h against a buffer (100 mM sodium phosphate, pH 7.5) containing 5% (*v*/*v*) acetonitrile at 4 °C. After an additional 24 h of dialysis against the same buffer in the absence of acetonitrile, fractions containing mOBP5 were analyzed using SDS–PAGE, pooled, and stored at −20 °C. All buffers used for purification were prepared with HLPC-grade water.

### 2.2. Recombinant mOBP5 Characterization

mOBP5 purity was assessed using SDS–PAGE, and its concentration was determined using spectrophotometry using its extinction coefficient measured at 280 nm (ε_280_ = 16,305 M^−1^.cm^−1^). The oligomeric state of mOBP5 was analyzed using size exclusion chromatography with a 24-mL bed volume Superdex 75 10/300 GL column (GE Healthcare) at room temperature. The column was equilibrated in a solution containing 100 mM potassium phosphate and 150 mM NaCl (pH 7.5) at 1 mL/min. Bovine serum albumin (67 kDa), chicken egg ovalbumin (43 kDa), bovine ribonuclease A (13.7 kDa), and aprotinine (6.5 kDa) purchased from Sigma, were employed as standards.

Circular dichroism (CD) spectra were recorded as previously described [42] using a JASCO J-815 spectropolarimeter (Jasco, Tokyo, Japan) equipped with a Peltier temperature-control system (JASCO MPTC-490S). CD spectra were recorded at 20 °C using a 0.01 cm thick quartz cell between 180 and 250 nm at 0.5 nm intervals with a 50 nm/min scan speed. Data were averaged over 10 accumulated scans. The buffer contributions were subtracted, and the data were converted to mean residue ellipticity in deg.cm^2^.dmol^−1^. Secondary structure proportions were computed using the deconvolution CDSSTR algorithm of DICHROWEB (http://www.cryst.bbk.ac.uk/cdweb/html; accessed on 1 July 2022).

### 2.3. Fluorescence-Based Binding Assay

Competitive ligand binding experiments were performed using N-phenyl-1-naphthylamine (NPN) as a fluorescent probe as previously described [42]. All measurements were conducted at 20 °C using a Cary Eclipse spectrofluorimeter (Agilent Technologies, Santa Clara, CA, United States) equipped with a Peltier temperature-control unit with a 1 cm light path, quartz cuvette, and 5 nm slits for both excitation and emission. To determine the binding affinity of mOBP5 for NPN, a 2 µM protein sample in 50 mM sodium phosphate buffer at pH 7.5 was titrated with aliquots of 1 mM NPN in 10% methanol to a final concentration of 0.1–25 µM. The excitation wavelength of NPN was 337 nm, and the fluorescence emission was recorded between 350 and 600 nm. The value of the dissociation constant (*K*_d_) NPN was calculated using SigmaPlot 12.5 software (Systat Software, Inc., San Jose, CA, USA), as previously described [42].

Competitive binding assays were carried out with 2 μM mOBP5 incubated with 10 μM NPN, as previously described [42]. After the addition of the competitor compound and the stabilization of the signal (3–4 min), the fluorescence spectrum was recorded. The IC_50_ value is the concentration of odorant, which causes a fluorescence decrease in half-maximal intensity. The IC_50_ values were calculated using SigmaPlot software. Dissociation constants (*K*_d_) for odorant competitors were calculated as *K*_d_ = [IC_50_]/(1 + [L]/*K*_d_ NPN), with [L] being the free fluorophore concentration and *K*_d_ being the OBP-fluorophore complex dissociation constant. The reported *K*_d_ values are the average of three measurements performed in three independent experiments.

### 2.4. Isothermal Titration Microcalorimetry (ITC)

Isothermal titration microcalorimetry (ITC) experiments were carried out at 25 °C with a VP-ITC microcalorimeter (Malvern Panalytical, Palaiseau, France) as previously described [34,42]. Before initiating the titration run, mOBP5 solutions (25 μM in 50 mM phosphate buffer, pH 7.5) were thoroughly degassed under vacuum. Ligand solutions (250 μM in 50 mM phosphate buffer, pH 7.5) were injected into one 3 µL aliquot, then 26 successive 10 μL aliquots. The duration of each injection was 20 s with a 210 s recovery time between injections. Heats of dilution determined in the absence of mOBP5 were subtracted from the titration data prior to curve fitting. Thermodynamic parameters were obtained by fitting the values to a single site binding model through nonlinear regression using Microcal Origin^®^ software.

### 2.5. Molecular Modeling and Molecular Docking

The protein model was obtained using Alpha Fold 2 [43]. All parameters were set to defaults, except for the number of recycling (an iterative process of reinjecting the output as a new template, Alpha Fold model improves accuracy of the predicted structure), which was set to 50. Twenty-five models were produced, and the model with the best score provided by Alpha Fold was selected. Protonation states were defined using PropKa [44,45] at pH 7.5. Nonpolar hydrogens were merged using AutoDock Tools [46].

Three-dimensional coordinates for the ligands were generated with GypsumDL [47]. Then, 1000 conformers per molecule were produced at pH 7.5, and the lowest energy conformer was selected. Partial charges were assigned to each atom, and all bonds were set to be rotatable with AutoDock Tools. Finally, molecular properties were obtained using RDKit 2022 [48].

Binding mode and interactions of the selected ligands with mOBP5 were performed using AutoDock Vina [49] software with default parameters. No prior protein conformational search was performed due to the rigidity of the OBP upon ligand binding [6]. The docking site on the protein was defined by establishing a grid box with dimensions of X: 17 Y: 17 Z: 15 Å centered on the binding cavity. Interactions of the ligand–protein complex were studied using LigPlot+ [50]. The theoretical CD spectrum was estimated using SESCA [51]. The secondary structure of the OBP was estimated using sequence information from the HBSS methodology. Alignment of the OBP sequence was performed using Jalview [52]. All 37 mammalian OBP sequences were obtained using the UniProt database [53]. The MAFFT [54] algorithm and its default parameters were used to align the sequences.

## 3. Results

### 3.1. Expression and Purification of mOBP5

To study the structure–function relationship of mOBP5, we heterologously expressed it in a prokaryotic system, as previously reported for pig OBP1 and rat OBP3 [42]. Using *E. coli* M15 [pREP4] cells, mOBP5 was expressed in high quantities as a soluble protein. SDS–PAGE analysis of the total cell lysate revealed that culture induced with 0.2 mM IPTG resulted in overexpression of mOBP5 with a molecular weight of approximately 20 kDa (Figure 1A). Vertebrate OBPs are difficult to obtain free of endogenous ligands because of their micromolar binding affinities and their ability to bind a broad range of ligands. Thus, natural and recombinant bovine OBPs have been cocrystallized with serendipitous ligands [55,56]. For all these reasons, mOBP5 was expressed as a His-tagged protein and was purified using immobilized metal-affinity chromatography using a recently developed protocol to remove potentially bound hydrophobic ligands [42]. SDS–PAGE (Figure 1B) and Western blot (Figure 1C) analysis of purified mOBP5 revealed >95% purity of the recombinant protein with a molecular mass of approximately 18 kDa, in agreement with the theoretical molecular mass value (18.4 kDa) of the mature protein. From 5 L of bacterial culture, we obtained approximately 26.4 mg of purified mOBP5.

### 3.2. Characterization of mOBP5

We used circular dichroism (CD) spectroscopy to confirm the correct folding of mOBP5. The far-UV CD spectrum of mOBP5 displayed a maximum at 196 nm and a minimum at 217 nm, which are characteristic of a protein containing a high proportion of beta-sheet secondary structures (Figure 2A). The deconvolution of the CD spectrum indeed revealed that mOBP5 was composed of an average of 48% beta-sheets and 4% alpha-helices, as expected for vertebrate OBPs. Given that mOBP5 has been suggested to form a heterodimer with mouse OBP7 (formerly known as mOBP1b) [24], we tested the oligomeric state of mOBP5. As shown in Figure 2B, calibrated exclusion-diffusion chromatography (Appendix A) of purified mOBP5 at 0.5 mg/mL exhibited an apparent molecular mass of 34 kDa. This value is close to the dimer theoretical mass value, strongly suggesting that mOBP5 acts mainly as a homodimer in solution at the tested concentration. In addition, the chromatogram revealed a broad peak with a leading edge, suggesting that mOBP5 may also exist as a trimer and/or tetramer.

### 3.3. Ligand Binding Properties of mOBP5

To investigate the binding properties of mOBP5, we first tested the ability of mOBP5 to bind the fluorescent reporter N-phenyl-1-naphthylamine (NPN). We found that the titration of mOBP5 with NPN was saturable, leading to a dissociation constant value of 3.33 ± 0.17 μM (Figure 3A). We then measured the displacement of NPN by 40 diverse odorant compounds, representing several classes of chemical structures and odors (Figure 3B). As illustrated by competitive curves (Figure 3C), the fluorescence intensity of the mOBP5-NPN complex was drastically reduced in the presence of various odorant molecules belonging to various chemical classes, such as alcohols (geraniol, 1-octanol, linalool, and menthol), aldehydes (octanal and lilialdehyde), ketones (menthone, 2-decanone, and (+)- and (−)-carvone) and the terpene compound, alpha-pinene, suggesting that NPN binds mOBP5 with a low level of non-specific binding. The binding affinities (K_d_ values) calculated from the half-maximal values (IC_50_ values) (Figure 3B and Appendix A) revealed that mOBP5 is a broadly tuned OBP with an affinity in less than the micromolar range for more active odorants. We found that mOBP5 had a weak binding capacity or no binding capacity for pyrazine and pyrazine derivatives. This poor binding could be due to the low solubility of these chemical compounds. We observed with some odorants an increase in fluorescence at high concentrations (Figure 3C), probably due to encapsulation of NPN in odorant micelles, as previously reported [14].

Binding experiments were further conducted using isothermal-titration microcalorimetry (ITC). This label-free technique is a direct method for assessing ligand–protein interactions, measuring the heat generated or absorbed during the binding reaction. ITC has been successfully used to study the binding process of vertebrate OBPs [17,34,57,58,59,60]. Six odorants, geraniol, octanol, citral, (+)-carvone, menthone, and isoamyl acetate, were studied. The thermograms (Figure 4) confirmed that binding onto mOBP5 was saturable showing the presence of approximately one binding site per monomer (Appendix A). The negative peaks revealed an exothermic interaction. The fitted thermodynamic parameters are listed in Appendix A. The dissociation constants (*K*_d_) were measured in the low micromolar range for all the tested odorants and were in reasonable agreement with binding affinities obtained using fluorescent probe displacement. Notably, the affinity of mOBP5 for odorants measured by ITC was higher than that measured using fluorescence displacement.

### 3.4. Docking Reveals Key Residues for mOBP5 Interactions

Molecular docking allows us to estimate both the orientation and the affinity of a ligand within a given protein [61]. Due to the lack of an experimental structure, a model of mOBP5 was generated using AlphaFold. The model contained 50% beta-sheets and 15% alpha-helices (9% of 4-helix and 6% of 3-helix). The theoretical CD spectrum was in good agreement with the experimental CD spectrum (Appendix A). The odorants were ranked according to their docking score [49], which is an estimation of binding affinity. In our case, we observed a significant correlation between the experimental ranking and docking score (Figure 5B) with a Spearman correlation of 0.6 (*p* value < 9 × 10^−5^). The 10 top ranked compounds were mostly binders, with two false positives (Figure 5B). In general, the docking score accurately discriminated between binders and nonbinders with an AUC value of 0.81 (Appendix A). Thus, based on the docking results, we identified the key residues from the mOBP5 binding site involved in ligand recognition (Figure 5A). They showed a rather low conservation, with the exception of PHE107. Most of them were hydrophobic [62] (Table 1), in agreement with the hydrophobic nature of most of the odorants.

More specifically, all mOBP5-binding complexes exhibited conserved hydrophobic interactions with five residues: TYR95, PHE107, VAL59, PHE123, and ILE43. The cavity also contained a hydrophilic patch composed of three amino acids, THR87, ASN93, and SER109 (Figure 5A). Thus, a long linear saturated hydrocarbon chain with a hydrophilic function seems to best fit the cavity (Appendix A). The hydrophilic moieties stabilize the odorant through an H-bond with one of the three polar amino acid residues (THR87, ASN93, or SER109). The structural analysis suggested that this stabilization influenced the orientation of the odorant inside the binding pocket. In particular, residue SER109 is aligned with ASN108 from rOBP1. This position is thought to be involved in ligand interactions [63]. Sequence conservation across 37 mammalian OBPs revealed that this position, although not conserved across the different species, was mainly polar.

The screening results indicated that mOBP5 recognized a large spectrum of odorants from different chemical families (Figure 3B). Therefore, we searched for a combination of molecular properties that could discriminate binders from nonbinders. We selected several physico-chemical descriptors according to their weight in a linear regression (*K*_d_ =−2.75 SASA+2.17 Volume+0.43 Docking Score with a corresponding r^2^ = 0.76). Correlation analysis between molecular descriptors and binding affinity (*K*_d_) revealed that solvent accessible surface area (SASA) and volume are the key factors controlling binding. We observed a correlation between SASA and volume with *K*_d_ values of 0.82 and 0.76, respectively (Figure 5C). To better understand the interaction between the molecules and OBP, molecular docking experiments were performed.

## 4. Discussion

We heterologously expressed mOBP5 in bacteria in high quantities as a soluble protein. Experimental CD showed that recombinant mOBP5 was folded into a secondary structure expected for a lipocalin and in agreement with those of the molecular model. We investigated its quaternary structure and found that mOBP5 behaves mainly as a dimer at neutral pH, as described for bovine OBP [64]. Since some proteins, such as OBPs, may undergo nonideal column interactions, the mOBP5 dimeric state should be confirmed using other methods, such as size exclusion chromatography-multiangle light scattering analysis. As already observed for other vertebrate OBPs [6,14,15,16,17], we found that mOBP5 is a broadly tuned OBP able to bind with a low micromolar affinity to a wide range of odorants belonging to different chemical classes. ITC experiments have shown that the binding stoichiometry of mOBP5 is below one site per monomer, indicating that the purification protocol [42] is not efficient enough to lead to a completely ligand-free protein. The *K*_d_ values determined using fluorescence displacements are higher than those measured using ITC experiments. This disparity can be accounted for by the failure to consider the non-specific binding of NPN in the fluorescence competitive assay.

Previous papers have investigated the influence of PTMs. These studies proposed that OBPs have an active role in odour detection, in particular, as a pheromone carrier. Specifically, the authors showed that phosphorylation would affect the recognition of biologically relevant ligands by porcine OBP. In particular, these modifications could abolish the recognition of two molecules involved in the pig species chemical communication: testosterone and palmitic acid. However, this type of PTMs would only have a minor influence on activity of other ligands and would induce variations of less than an order of magnitude on affinity [20]. Other PTMs (*O*-GLcNAcylation) were also proposed to play a role in pheromone perception [18]. It is worth noting that the two identified *O*-GlcNAc sites from pig OBP (SER13 and SER19) are located far away from the binding cavity. Thus, the mechanism by which the sugar would influence the ligand binding remains unknown. It thus appears that PTMs would not have any major impact on our results as we mostly focused our study on molecules which are not identified as pheromones. Sequence analysis using NetNGlyc - 1.0 software predicted the presence of one *N*-glycosylation site in mOBP5 (ASN101). Since bacteria do not realize glycosylation, it would be interesting to express mOBP5 using a eucaryotic expression system capable to achieve *N*-glycosylation, such as the yeast *Pichia pastoris,* and evaluate the influence of glycosylation in mOBP5 binding properties.

Binding experiments have revealed that mOBP5 is a broadly tuned OBP with a low affinity for pyrazine and pyrazine derivatives. As bovine OBP was originally identified as pyrazine binder protein, it is thus surprising to find one OBP which shows no or low affinity for this class of molecules [65]. It has been reported that among the three rat OBPs, only rat OBP3 has micromolar affinity for the two pyrazine derivatives studied here [15]. From a physiological point of view, pyrazine containing molecules are found in the wolf urine (a mouse predator) and are described as kairomones that induce a freezing or avoidance behaviour [66]. Thus, our results suggest that the detection of these compounds should involve another mouse OBP. Interestingly, transcriptomic analysis has revealed sexual dimorphism in mouse OBP expression [25]. Thus, mOBP5 is higher expressed in females that in males. This sexual difference and expression in lachrymal glands [29] suggest that mOBP5 can be involved in the detection of pheromones.

Molecular modeling combined with docking showed that the binding of mOBP5 to odorants is mainly governed by the hydrophobicity and volume of the odorants. This is in line with a previous large-scale study which showed that hydrophobicity alone could not explain the specificity of OBP [65]. This aligns well with the general binding behavior of OBPs with a broad spectrum of recognition [13]. The analysis of the binding cavity properties of the available crystallographic structure revealed no significant difference in terms of volume, polarity, or hydrophobicity among bovine OBP:1OBP [67]; rat OBP3:3ZQ3 [63]; rat OBP1:3FIQ [68]; pig OBP:1A3Y [69]; panda OBP3:5NGH [14]; and the mOBP5 model (Appendix A). None of these OBPs were described as specific to a chemical family of odorants. On the other hand, the cavity of human OBPIIa [70] appeared to be more polar and larger than the other cavity, with a higher affinity for aldehydic compounds [10]. However, even though OBPs bind to different chemical families, Löbel and collaborators [15] showed that different subtypes of OBPs recognize a complementary set of ligands.

## 5. Conclusions

In this paper, we combined experimental and numerical methods to investigate the binding of odorants to mOBP5. As observed for other vertebrate OBPs, we found that mOBP5 has a broad recognition spectrum. We observed that its binding affinity is mainly governed by ligand size. The agreement between docking and experimental ranking shows that numerical methods are promising tools to identify new ligands for mOBP5. Because OBPs are stable and easy to produce in large quantities at low cost, they have great potential for use as sensing elements in biosensors [34,36,37,38]. Our data provide a new OBP with known binding properties that might be useful for the design of electronic noses.

## Figures and Tables

**Figure 1 biology-12-00002-f001:**
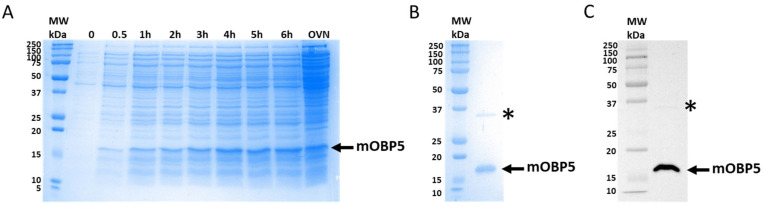
Electrophoretic analysis of mOBP5 expressed in bacteria. (**A**) *E. coli* M15 [pREP4] strain containing the pQE31-mOBP5 vector before and after IPTG induction for 30 min, 1 h, 2 h, 3 h, 4 h, 5 h, 6 h, and overnight (OVN). The position of mOBP5 is shown with an arrow. (**B**) SDS–PAGE of purified mOBP5 using immobilized metal-affinity chromatography. The proteins were analyzed using 12% SDS–PAGE and detected using Coomassie blue staining. The molecular weight markers (kDa) are shown. The asterisk indicates a band probably corresponding to an SDS-resistant dimer of mOBP5, as demonstrated with Western blot analysis. (**C**) Western blot analysis of mOBP5. The purified protein from the gel filtration was separated using SDS–PAGE followed by Western blotting using mouse anti-HIS primary antibody and goat anti-mouse horseradish peroxidase conjugated secondary antibody. The arrow indicates the position of purified mOBP5. The asterisk indicates a band probably corresponding to an SDS-resistant dimer of mOBP5.

**Figure 2 biology-12-00002-f002:**
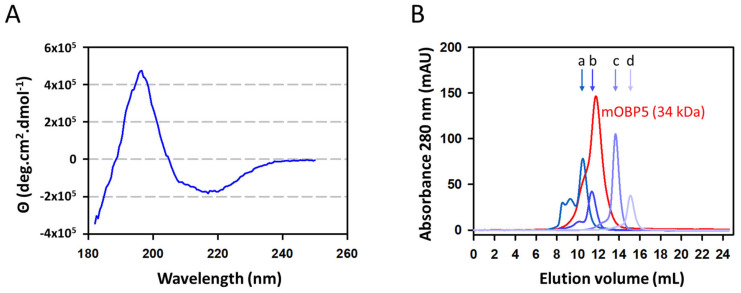
Secondary structure analysis and molecular weight estimation of mOBP5. (**A**) Circular dichroism analysis of mOBP5. The far-UV spectrum was recorded in 50 mM NaF buffer at pH 7.5 with 15 μM protein using a cuvette path length of 0.1 mm. (**B**) Exclusion-diffusion chromatography on a Superdex 75 column of mOBP5. The elution positions of the molecular mass standards are indicated by arrows: a, bovine serum albumin (67 kDa); b, chicken egg ovalbumin (43 kDa); c, ribonuclease (13.7 kDa); d, aprotinine (6.5 kDa).

**Figure 3 biology-12-00002-f003:**
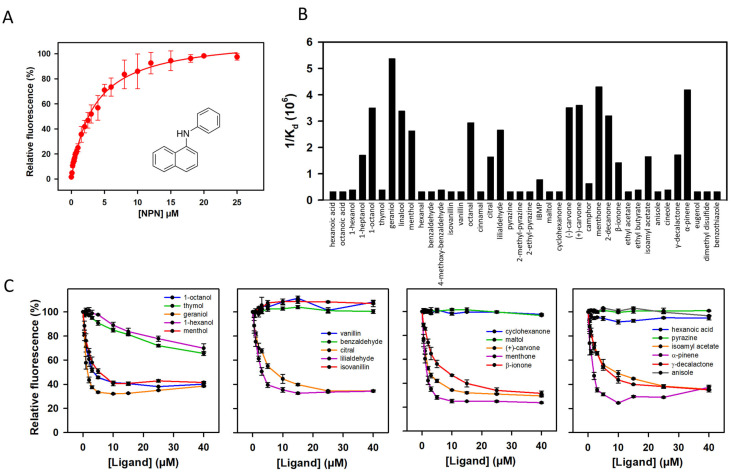
Binding properties of mOBP5 measured using a competitive fluorescent assay. (**A**) Titration curve of mOBP5 with NPN; circles show experimental data, while the solid lines are the computed binding curves; excitation and emission wavelengths were 337 and 420 nm, respectively; the mOBP5 concentration was 2 µM. The NPN structure is inserted. (**B**) Affinity of mOBP5 toward the 40 tested odorants clustered by chemical families. (**C**) Examples of competitive binding curves toward NPN with several odorants. The fluorescence of the NPN-mOBP5 complex was designated as 100% in the absence of a competitor.

**Figure 4 biology-12-00002-f004:**
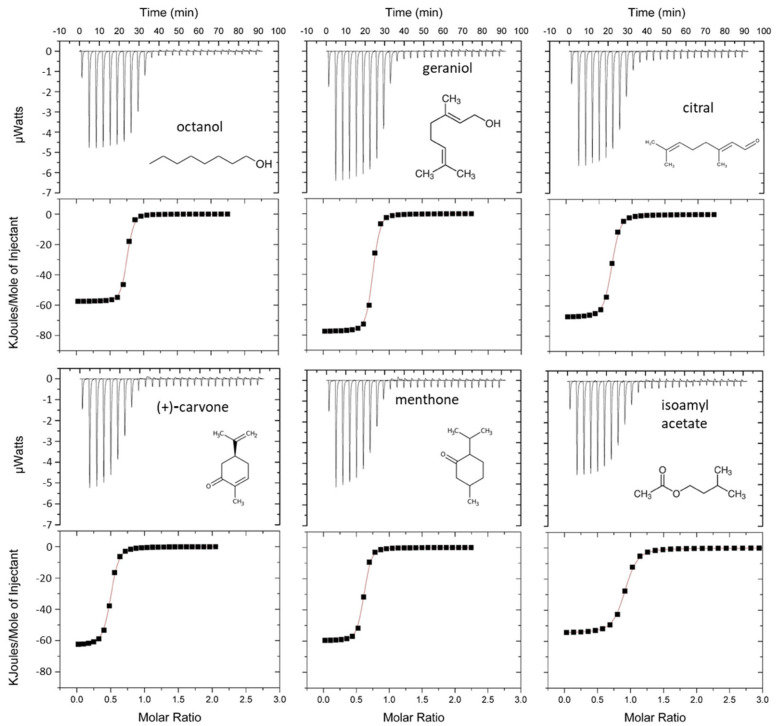
Binding properties of mOBP5 measured using isothermal titration calorimetry. The experiments have been carried out at 25 °C, in 100 mM phosphate buffer, pH 7.5. The concentrations of the reactants were 250 µM odorant and 25 μM mOBP5. The top panels are the thermograms, and the bottom panels are the fitted binding isotherms. The structures of the odorants are shown.

**Figure 5 biology-12-00002-f005:**
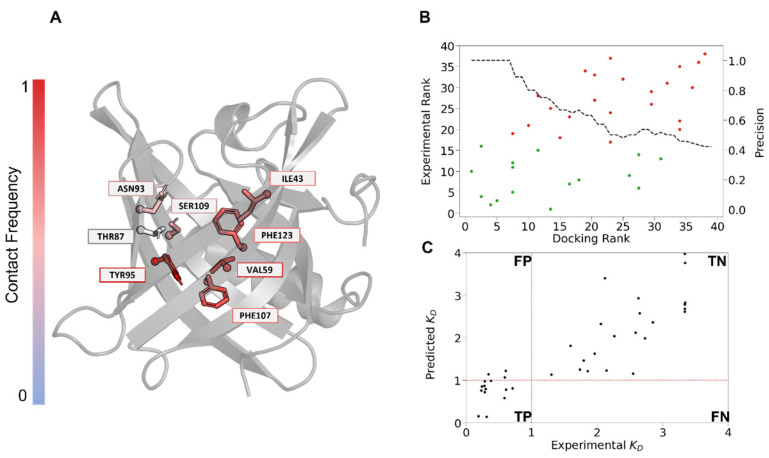
(**A**) Binding pocket residues of mOBP5, colored according to their frequency of contact. The color scale ranges from red, as the maximum frequency, to blue, as the minimum frequency. Only the sidechain for each amino acid residue is represented, and the alpha carbon appears as a sphere. (**B**) Evaluation of the performance of the docking protocol in terms of the ranking of odorant to mOBP5. Dots in green and red account for binders and nonbinders, respectively. The dotted line represents precision for each rank defined as the threshold. (**C**) Linear regression analysis of binding affinity (*K*_d_) using docking score, volume, and SASA as descriptors. The dotted red line defines the confusion matrix with False Positive (FP), True Positive (TP), False Negative (FN), and True Negative (TN).

**Table 1 biology-12-00002-t001:** Main residues in the binding site. The frequency is calculated using the ratio of contact between the residue and the ligand for all poses obtained for the agonists (16 binders). Conservation was calculated as the proportion of all homologous sequences containing this residue after alignment.

Residue	Interaction	Frequency	Conservation	Conservation Type
TYR95	Hydrophobic	99%	3%	Hydrophobic
VAL59	Hydrophobic	91%	11%	Hydrophobic
PHE107	Hydrophobic	85%	61%	Aromatic
PHE123	Hydrophobic	80%	3%	Polar
ILE43	Hydrophobic	77%	5%	-
ASN93	H-Bond	67%	11%	Polar
SER109	H-Bond	65%	5%	Small
THR87	H-Bond	48%	34%	-

## Data Availability

Not applicable.

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
