# Peer review of "Ligand Binding Properties of Odorant-Binding Protein OBP5 from Mus musculus"

_biology, 2022, doi:10.3390/biology12010002_

Round 1

Reviewer 1 Report

The study by Moitrier et al functionally characterized OBP5 from Mus musculus. OBP5 was recombinantly expressed in bacteria and the recombinant protein was used for the fluorescent competitive binding assay to investigate the binding affinity for 40 odorants. The results show that OBP5 is a broadly tuned OBPs with good binding to alcohols, aldehydes, ketones, and more. Moreover, the structural modeling work is also a good try. The conclusion is supported by the data. The manuscript is well written. I have some minor comments below:

Line 70. It is better to introduce in detail the expression pattern of these OBPs in mice. Of these four OBPs, which one is most highly expressed? Where are they located?

In the introduction, the authors should explain why they choose OBP5 other than others to study.

As far as I know, pyrazine is very difficult to dissolve in solvent. Could this be the reason for the poor binding of OBP5 to this compound?

The Discussion is too limited. I suggest an extrapolation of what OBP5 does in mice olfaction based on the binding data.

All in all, it is a good piece of work on OBPs.

Author Response

Comments and Suggestions for Authors

The study by Moitrier et al functionally characterized OBP5 from Mus musculus. OBP5 was recombinantly expressed in bacteria and the recombinant protein was used for the fluorescent competitive binding assay to investigate the binding affinity for 40 odorants. The results show that OBP5 is a broadly tuned OBPs with good binding to alcohols, aldehydes, ketones, and more. Moreover, the structural modeling work is also a good try. The conclusion is supported by the data. The manuscript is well written. I have some minor comments below:

Line 70. It is better to introduce in detail the expression pattern of these OBPs in mice. Of these four OBPs, which one is most highly expressed? Where are they located?

Our reply: We thank the reviewer for his/her positive comments on our study. We provide now in the Introduction part more details in the text about the expression pattern of these OBPs and where they are located. We clarified which one is the most highly expressed.  See Line 72: “At the RNA level, OBP2, OBP1, and OBP5 are the most strongly expressed OBPs in the main olfactory epithelium [22] in agreement with protein isolation from mouse nasal tissue [24]. In situ hybridization experiments have demonstrated that genes encoding mouse OBP5 and OBP7 are expressed in the septal and in the lateral nasal glands, but not in glands from the olfactory mucosa [24,26]. Although the functional role of OBPs is not clearly understood, using a fluorescently labeled mOBP5 loaded with an odorant, Strotmann and Breer have showed a rapid internalization of OBP inside the sustentacular cells of the mouse epithelium [27].”.

In the introduction, the authors should explain why they choose OBP5 other than others to study.

Our reply: We clarified this point and modified this sentence in the Introduction part, Line 93: “Here, we report the expression of functional mouse mOBP5 using Escherichia coli, which is one of the most abundantly expressed OBPs in olfactory tissues of house mouse [22,24].”.

As far as I know, pyrazine is very difficult to dissolve in solvent. Could this be the reason for the poor binding of OBP5 to this compound?

Our reply:  We are not convinced that the lack of pyrazine binding to OBP comes from a solubility issue since some OBPs binds pyrazine and pyrazine derivatives with great affinities. We added a special focus about pyrazine in the discussion section and added this paragraph, see Line 263: “This poor binding could be due to the low solubility of these chemical compounds. We observed with some odorants an increase of fluorescence at high concentrations (Figure 3C), probably due to encapsulation of NPN in odorant micelles, as previously reported [14].”

The Discussion is too limited. I suggest an extrapolation of what OBP5 does in mice olfaction based on the binding data.

Our reply: We took into account the reviewers comment and added a paragraph on the putative role of OBPs in the recognition of odorants in the discussion part. See Line 387: “Binding experiments have revealed that mOBP5 is a broadly tuned OBP with a low affinity for pyrazine and pyrazine derivatives. Bovine OBP was originally identified as pyrazine binder protein, it is thus surprising to find one OBP, which shows no or low affinity for this class of molecules [65]. It has been reported that among the three rat OBPs, only rat OBP3 has micromolar affinity for the two pyrazine derivatives studied here [15]. From a physiological point of view, pyrazine containing molecules are found in the wolf urine (a mouse predator) and are described as kairomones that induce a freezing or avoidance behaviour [66]. Thus, our results suggest that the detection of these compounds should involve another mouse OBPs. Interestingly, transcriptomic analysis has revealed sexual dimorphism in mouse OBP expression [25]. Thus, mOBP5 is higher expressed in females that in males. This sexual difference and expression in lachrymal glands [29] suggest that mOBP5 can be involved in the detection of pheromones.”.

All in all, it is a good piece of work on OBPs.

Reviewer 2 Report

The paper entitled “Ligand binding properties of odorant-binding protein OBP5 2 from Mus musculus” reports the functional characterization of the mouse Odorant Binding Protein OBP5 (mOBP5), also known as OBP1a. To this aim, the protein has been recombinantly expressed as a hexahistidine-tagged protein in bacteria and purified. The oligomeric state and secondary structure composition of mOBP5 were investigated. The binding properties of the recombinant OBP5 was studied with 40 odorants. The study shows that mOBP5 binds a restricted 34 number of odorants with good affinity. Finally, the team determines which amino acid residues of the mOBP5 share its binding properties.

The manuscript is clear and it is well structured and presented. The references cited are relevant. All the figures and tables are appropriate, properly show the data. The field of research is important and increasing the knowledge in the Odorant binding proteins is of interest for biotechnology. They can be used for the development of sensing elements in biosensors for odor monitoring, clinical analysis or food sensory evaluation. OBPs can also be an efficient solution to prevent and/or remove unpleasant odorant molecules trapped on the surface of textiles.

Nevertheless, I would like to address two major points that must at least be discussed in the paper:

1.     The mOBP5 exhibits one N-glycosylation site. The glycosylation of the protein can change its binding properties as mentioned in the references by the authors. Bacteria do not realize glycosylation. The hypothesis of the influence of the glycosylation cannot be tested. It's unfortunate. Authors do not even discuss of this point. If the protein is not recombinantly expressed in another expression system such as yeast (Pichia pastoris) able to glycosylate proteins, authors should at least tone down their conclusions concerning the binding properties of the mOBP5. Their conclusion must be restricted to the non-glycosylated recombinant protein, until the presence of a sugar on Asn101 will be tested on its binding properties. Moreover, the glycosylation site Asn101, is not far from the amino acid residues designated involved in binding: Asn63, Tyr95, Phe107 and Ser109. This reinforces and underlines the importance to design an experimental and appropriate expression of the recombinant protein to test the influence of the glycosylation in the binding properties.

At least, this must be discussed or the conclusions must be restricted to the recombinant form studied.

2.     The second comment concerns the native mOBP5. The recombinant OBP5 obtained should be compared to the native form of the protein. Authors should extract proteins from the olfactory mucosa or mucus and should add a Coomassie blue of the total proteins extracted and separated by electrophoresis and authors should provide also a western blot analysis of the native mOBP5 extracted from the nose. The results should be added in figure1 (Coomassie blue) and in figure S2 (western blot). Moreover, the western blots could be added in figure 1 and not as supplementary figure.

Author Response

Comments and Suggestions for Authors

The paper entitled “Ligand binding properties of odorant-binding protein OBP5 from Mus musculus” reports the functional characterization of the mouse Odorant Binding Protein OBP5 (mOBP5), also known as OBP1a. To this aim, the protein has been recombinantly expressed as a hexahistidine-tagged protein in bacteria and purified. The oligomeric state and secondary structure composition of mOBP5 were investigated. The binding properties of the recombinant OBP5 was studied with 40 odorants. The study shows that mOBP5 binds a restricted 34 number of odorants with good affinity. Finally, the team determines which amino acid residues of the mOBP5 share its binding properties.

The manuscript is clear and it is well structured and presented. The references cited are relevant. All the figures and tables are appropriate, properly show the data. The field of research is important and increasing the knowledge in the Odorant binding proteins is of interest for biotechnology. They can be used for the development of sensing elements in biosensors for odor monitoring, clinical analysis or food sensory evaluation. OBPs can also be an efficient solution to prevent and/or remove unpleasant odorant molecules trapped on the surface of textiles.

Our reply: We thank the reviewer for his/her insightful remarks and suggestions.

Nevertheless, I would like to address two major points that must at least be discussed in the paper:

  1. The mOBP5 exhibits one N-glycosylation site. The glycosylation of the protein can change its binding properties as mentioned in the references by the authors. Bacteria do not realize glycosylation. The hypothesis of the influence of the glycosylation cannot be tested. It's unfortunate. Authors do not even discuss of this point. If the protein is not recombinantly expressed in another expression system such as yeast (Pichia pastoris) able to glycosylate proteins, authors should at least tone down their conclusions concerning the binding properties of the mOBP5. Their conclusion must be restricted to the non-glycosylated recombinant protein, until the presence of a sugar on Asn101 will be tested on its binding properties. Moreover, the glycosylation site Asn101, is not far from the amino acid residues designated involved in binding: Asn63, Tyr95, Phe107 and Ser109. This reinforces and underlines the importance to design an experimental and appropriate expression of the recombinant protein to test the influence of the glycosylation in the binding properties.

At least, this must be discussed or the conclusions must be restricted to the recombinant form studied.

Our reply: While ASN101, and overall N-glycosylation motif (NxS/T), is not a conserved position across mammals OBPs (11%), this position is also not consistent with spatial localization of this site with those reported in the literature. Finally, this position is hindered by mOBP5 alpha helix, thus this site unlikely being a N-glycosylation site.

Nevertheless, we took into account the reviewers comment and added a paragraph in the discussion section, see Line 369: “Previous papers have investigated the influence of PTMs. These studies proposed that OBPs have an active role in odour detection, in particular as a pheromone carrier. Specifically, the authors showed that phosphorylation would affect the recognition of biologically relevant ligands by porcine OBP. In particular, these modifications could abolish the recognition of two molecules involved in the pig species chemical communication: testosterone and palmitic acid. However, this type of PTMs would only have a minor influence on activity of other ligands and would induce variations of less than an order of magnitude on affinity [20]. Other PTMs (O-GLcNAcylation) were also proposed to play a role in pheromone perception [18]. It is worth to note that the two identified O-GlcNAc sites from pig OBP (SER13 and SER19) are located far away from the binding cavity. Thus, the mechanism by which the sugar would influence the ligand binding remains unknown. It thus appears that PTMs would not have any major impact on our results as we mostly focus our study on molecules which are not identified as pheromone. Sequence analysis using NetNGlyc - 1.0 software predicted the presence of one N-glycosylation site in mOBP5 (ASN101). Since bacteria do not realize glycosylation, it would be interesting to express mOBP5 using an eucaryotic expression system capable to achieve N-glycosylation such as the yeast Pichia pastoris and evaluate the influence of glycosylation in mOBP5 binding properties.”.

  1. The second comment concerns the native mOBP5. The recombinant OBP5 obtained should be compared to the native form of the protein. Authors should extract proteins from the olfactory mucosa or mucus and should add a Coomassie blue of the total proteins extracted and separated by electrophoresis and authors should provide also a western blot analysis of the native mOBP5 extracted from the nose. The results should be added in figure1 (Coomassie blue) and in figure S2 (western blot). Moreover, the western blots could be added in figure 1 and not as supplementary figure.

Our reply: We believe that our story could be extended, but the goal of this paper was based on a recombinant expression of mOBP5. We agree that more experiments could be carried out on native mOBP5 extracted from the nose but these experiments would require a lot more effort and much more time to be conclusive. In our study, we used anti-histag antibodies to detect mOBP5. To detect native mOBP5, we would need to generate specific antibodies directed against native mOBP5.

As suggested by the referee, the western blot initially presented in the supplementary data has been added in the Figure 1 as Figure 1C.

Author Response

Comments and Suggestions for Authors

Moitrier L. et al. ‘Ligand binding properties of odorant-binding protein OBP5 from Mus musculus’

The authors report the expression in E. coli of a murine odorant-binding protein and its partial characterisation. The function in vivo of such proteins is not known, but they are believed to have a role in the transport of non-polar odorant molecules to olfactory receptors. The authors use their previous methodological approach for OBPs from other species (Ref. 41), so in this sense the manuscript is not fully original, but it does provide information on a further OBP. The rationale for choosing the potential ligands is not described, but a significant number (40) is assessed and these cover a range of chemical classes.

Our reply: We thank the reviewer for this comment. We added the following sentence page 2, line 96-97:

“As the recognition spectrum of this OBP is unknown, molecules from different chemical families with different impact on behavior were selected [41].”

The authors point out in the introduction that binding specificity may be altered by post-translational modifications, but the protein used here is unmodified. The potential impact of post-translational modifications and their possible implications for the present findings should be discussed in the Discussion (which currently is rather brief).

Our reply: We took into account the reviewer comment and added a paragraph on the post-translational modifications in the discussion Part, see Line 369-386:

“Previous papers have investigated the influence of PTMs. These studies proposed that OBPs have an active role in odour detection, in particular as a pheromone carrier. Specifically, the authors showed that phosphorylation would affect the recognition of biologically relevant ligands by porcine OBP. In particular, these modifications could abolish the recognition of two molecules involved in the pig species chemical communication: testosterone and palmitic acid. However, this type of PTMs would only have a minor influence on activity of other ligands and would induce variations of less than an order of magnitude on affinity [20]. Other PTMs (O-GLcNAcylation) were also proposed to play a role in pheromone perception [18]. It is worth to note that the two identified O-GlcNAc sites from pig OBP (SER13 and SER19) are located far away from the binding cavity. Thus, the mechanism by which the sugar would influence the ligand binding remains unknown. It thus appears that PTMs would not have any major impact on our results as we mostly focus our study on molecules which are not identified as pheromone. Sequence analysis using NetNGlyc - 1.0 software predicted the presence of one N-glycosylation site in mOBP5 (ASN101). Since bacteria do not realize glycosylation, it would be interesting to express mOBP5 using an eucaryotic expression system capable to achieve N-glycosylation such as the yeast Pichia pastoris and evaluate the influence of glycosylation in mOBP5 binding properties.”

In this context it would be helpful if the CD study, rather than being just theoretical, were to include actual CD spectra of the bacterial-expressed protein and the mouse protein (if they have it) or mOBP5 expressed in a eukaryotic expression system.

Our reply: The CD spectrum of the recombinant mOBP5 expressed in bacteria is shown in Figure 2A.

Gel-filtration chromatography of purified mOBP5 reveals a broad peak with a leading edge, which suggests that not only is a dimer present, but also a trimer and/or tetramer. It is also not clear if the stoichiometry of less than one bound ligand relates to the monomer, dimer etc and this should be clarified.

Our reply: We agree with the referee’s remark. We added a paragraph in the result section, page 6, line 239-241 “In addition, the chromatogram revealed a broad peak with a leading edge, suggesting that mOBP5 may also exists also as a trimer and/or tetramer.”.

The authors need to explain why NPN shows no non-specific binding (Fig 3a), while all the active competitors do (Fig 3C). The Kd values for the binding ligands are in the low μM range, but the Kds calculated by the authors are higher when determined by fluorescence displacement than by ITC. My suspicion is that much of the disparity can be accounted for by the failure to allow for the non-specific binding in the fluorescence displacement graphs.

Our reply: We took into account the reviewer comment and added a paragraph to explain the non-specific binding and the disparity between Kd values obtained by fluorescence and ITC. We added this paragraph in the discussion section, page 10, line 365-368: “The Kd values determined by fluorescence displacements are higher than those measured by ITC experiments. This disparity can be accounted for by the failure to consider the non-specific binding of NPN in the fluorescence competitive assay.”.

Other points

Simple Summary (line 25): change to ‘potentially involved’

Our reply: The change has been made.

Abstract (line 37); change ‘revealed’ to ‘indicated’

Our reply: The change has been made.

Line 116: state the temperature of the chromatographic separation 

Our reply: We added the temperature. See Line 127.

Line 163: number of recycling?

Our reply: Recycling is a tuneable parameter introduced in the Alphafold2 model. Structure is predicted from the sequence and can be helped by template structure (Crystallographic structure sharing high sequence identity with the target). They also propose to introduce as the template the first output of one run of the model. Through an iterative process of reinjecting the output as a new template (or recycling), alphafold model improves accuracy of the predicted structure.

We added a sentence in the manuscript, Line 176-179: ” The protein model was obtained using Alpha Fold 2 [43]. All parameters were set to defaults, except for the number of recycling, (an iterative process of reinjecting the output as a new template, Alpha Fold model improves accuracy of the predicted structure), which was set to 50.”. (an iterative process of reinjecting the output as a new template, Alpha Fold model improves accuracy of the predicted structure).

Line 164: best score? For what?

Our reply: Alphafold model measures confidence of the predicted structure in terms of pIDDT metric as well as pTM metrics. Those scores had good correlations to actual prediction accuracy.

We added a precision in sentence line 180.

Twenty-five models were produced, and the model with the best score provided by Alpha Fold was selected.

Line 237: why is this ‘interesting’? Explain

Our reply: We added a small discussion about pyrazine and derivatives in the discussion, page 11, Line 387 “Binding experiments have revealed that mOBP5 is a broadly tuned OBP with a low affinity for pyrazine and pyrazine derivatives. Bovine OBP was originally identified as pyrazine binder protein, it is thus surprising to find one OBP, which shows no or low affinity for this class of molecules [65]. It has been reported that among the three rat OBPs, only rat OBP3 has micromolar affinity for the two pyrazine derivatives studied here [15]. From a physiological point of view, pyrazine containing molecules are found in the wolf urine (a mouse predator) and are described as kairomones that induce a freezing or avoidance behaviour [66]. Thus, our results suggest that the detection of these compounds should involve another mouse OBPs. Interestingly, transcriptomic analysis has revealed sexual dimorphism in mouse OBP expression [25]. Thus, mOBP5 is higher expressed in females that in males. This sexual difference and expression in lachrymal glands [29] suggest that mOBP5 can be involved in the detection of pheromones.”.

Fig 3b: what is the significance of the colour coding of the histogram bars?

Our reply: Thank you for this comment. The colours accounted for the chemical families. It has been added in the caption. To clarified this point, the histogram bars were coloured in black.

Line 266: 15% seems significantly larger than 4%

Our reply: We are thankful for the reviewer. We were combining the alpha helix percentage (9 %) with the percentage of 310helix (6%). The difference is the result of two more helical turn in the theoretical model. We modified the text. See Line 297.

Line 306: I do not understand what this sentence means scientifically

Our reply: This position (SER109) has been already identified has being crucial for ligand recognition in another OBP (rOBP1).

Table 1: if there were 40 ligands, how is frequency of 99% possible, as each accounts for 2.5% of the frequency?

Our reply: We are grateful for the careful reading from the reviewers. We were not mentioning that this table corresponds to the binding information on only binders (16 ligands). This precision has been added in the table caption.

The frequency is calculated on all the docking poses of the binders. Indeed, while having preferred orientation, OBP binders are shown to have a certain degree of freedom in the cavity. Taking all the docking pose allows to take in account potential other binding conformation. This is why we end with a frequency of 99%.

References: Refs 9 and 13 are the same

Our reply: We checked the references and removed the duplicates.

Fig S3: bovine serum albumin (no ‘e’)

Our reply: The change has been made.

Fig S7: docking experiments (plural)

Our reply: The change has been made.

Table S1: where the arbitrary value of 200 μM is used, the Kd values should be shown as >3.16; the three right-hand columns should be explained in the legend

Our reply: The changes have been made and the three right-hand columns have been explained in the legend.

Round 2

Reviewer 2 Report

The paper entitled “Ligand binding properties of odorant-binding protein OBP5 from Mus musculus” reports the functional characterization of the mouse Odorant Binding Protein OBP5 (mOBP5), also known as OBP1a. 

1- Authors add a paragraph line 369, discussing of the post translational modifications of the OBP using literature, and giving scientific soudness to the work they realized. 

2- As suggested, the western blot initially presented in the supplementary data has been added in the Figure 1 as Figure 1C.

3- I agree, with the authors, detecting the native mOBP5, need to generate specific antibodies directed against native mOBP5, that is to long tfor the present paper. As the antibodies against OBPs are not commercially availlable.

Thank you for the work and the nice answers.